# Sound-evoked pupil dilation quantifies misophonic symptoms

**Jan Willem de Gee** [1,2]*, **Laia Alonso-Marmelstein** [3], **Kate Schwarz-Roman** [3], **Romke Rouw** [2,3]*

**1** Swammerdam Institute for Life Sciences, University of Amsterdam, Amsterdam, The Netherlands, **2** Amsterdam Brain and Cognition, University of Amsterdam, Amsterdam, The Netherlands, **3** Department of Psychology, University of Amsterdam, Amsterdam, The Netherlands

* j.w.degee@uva.nl (JWG); r.rouw@uva.nl (RR)

## Abstract

Misophonia is a debilitating disorder where seemingly innocuous sounds (often human made, such as chewing or throat clearing) evoke intense negative cognitive, emotional and physical "fight-or-flight" responses. Recent studies reported alarmingly high prevalence across different countries and population characteristics, revealing an urgent need for a better understanding of this condition as well as improved measurement and diagnostic tools. First, current misophonia symptom measurements rely on (subjective) self-reports, and different studies employ various diagnostic approaches and cut-off scores. There is an urgent need for a complementary (objective) psychophysiological measurement tool. Second, the role of "mild" or "moderate" symptoms is currently topic of debate: are they still manifestations of the misophonic disorder? Here, we employ pupillometry to map out arousal responses to misophonia trigger sounds. We show that (i) pupil dilation can reliably differentiate misophonic responses from responses to generally unpleasant sounds (such as nails on chalk board), (ii) the "milder cases" of misophonia still show arousal responses characteristic of misophonia, and (iii) pupillometry can even be used to aid diagnosis in an individual; based on only pupil dilation, misophonic symptom severity could be reliably predicted at the level of a single individual. We conclude that even mild misophonic responses can reliably, objectively and cost-effectively be indexed by pupil-linked arousal.

## Introduction

Individuals with misophonia experience strong negative emotions like rage or disgust in response to everyday, often human-made, sounds (e.g., chewing or throat clearing) [1–6]. The ubiquitous nature of the "trigger" stimuli and strength of the misophonic response can lead to severe distress and even suicide ideation [7–9]. Misophonia is a newly defined disorder, quickly gaining scientific attention across

**Data availability statement:** All resources, including sounds, data and code used for the analyses in this paper (in Python), are publicly available at Zenodo: https://doi.org/10.5281/zenodo.18202830.

**Funding:** This research was supported by a grant from the Misophonia Research Fund (to J.W. de Gee and R. Rouw) and by a grant from the Dutch Research Council (Nederlandse Organisatie voor Wetenschappelijk Onderzoek; grant number, grant VI.Veni.232.210; to J.W. de Gee). The funders had no role in study design, data collection and analysis, decision to publish, or preparation of the manuscript.

**Competing interests:** The authors have declared that no competing interests exist.

scientific disciplines (e.g., audiology, cognitive science, clinical psychology, occupational therapy, psychiatry, and neuroscience) [3–5,7–15]. Despite rapidly advancing research, much remains unknown about the etiology, course, or mechanisms of the condition, leaving open the question why seemingly innocuous daily sounds would evoke such strong aversive ("fight or flight") responses.

The need for a better understanding of misophonia extends beyond fundamental scientific interest. A recent series of studies revealed alarmingly high misophonia prevalence numbers of ~1 in 5 in general (non-clinical) population samples across different nations (Turkey [16], UK [17], India [18], China [19], USA [20]). Results furthermore showed a wide range of symptom severity: "severe misophonia" was reported in 5% to 13% of the population [16,20–22] while the same studies also report "some" level of misophonic symptoms in as much as 79% of the population [16,20]. This reflects the current struggle on how to interpret "mild" or "moderate" symptoms: are they still manifestations of a misophonic response? Put differently, can "mild" misophonic responses be discerned from normal annoyance to unpleasant sounds? The call for improved misophonia diagnostic and measurement tools thus entails a need for methodology that reliably distinguishes misophonic from non-misophonic complaints, while simultaneously being sensitive to variations in misophonic response severity.

Misophonia research would benefit from a physiological tool to objectively characterize individual variation in misophonia symptom severity. To date, researchers have largely measured misophonic complaints using self-reports. As the field is quickly expanding, assessment approaches include questionnaires, structured diagnostic interviews and standardized experimental designs [7,23–25]. Yet, misophonia researchers have pointed at the limitations of employing only self-report measurements [23,25], which are prone to response bias: participants may underreport or exaggerate symptoms due to social desirability, memory errors, mood, or personal interpretations. Furthermore, while diagnostic psychiatric criteria have been formulated [3], different studies still employ various diagnostic approaches and different cut-off scores [6,7,10,24], complicating cross-study comparisons or conclusions.

Pupillometry is easy to use, cost-effective, and may be employed on a large scale to objectively and sensitively quantify misophonia severity through sound-evoked pupil responses: (i) trigger sounds elicit exaggerated responses in the anterior insula and anterior cingulate cortex [4,11], key regions of the cortical salience network, (ii) these structures send top-down projections to the subcortical ascending arousal network, including the noradrenergic locus coeruleus [26–28], (iii) ascending arousal nuclei, in turn, increase sympathetic activity (fight-or-flight) and suppresses parasympathetic activity (rest-and-digest) during stress, arousal, and attention [29,30], (v) pupil size fluctuations at constant luminance reflect activity of the cortical saliency network, neuromodulatory activity and autonomic arousal [27,28,31–35], and, as the above circuit would predict, (vi) pupil responses scale with the emotional intensity of auditory stimuli [36–38]. Other physiological measures, such as galvanic skin response and heart rate variability [1,4,11,39], suffer from practical issues in the context of misophonia: these methods are relatively noisy and sluggish [40] and require

"on-the-body" measurements (e.g., fitting equipment or on-skin glue), which is a significant drawback for clinical groups. Additionally, neuroimaging [4,11] is expensive and therefore unsuitable for large-scale diagnosis.

How pupillary response correlates with individual misophonic severity has not yet been studied. We hypothesized that the pupil would dilate in response to both generally unpleasant sounds and common misophonia trigger sounds, and, across individuals, the pupil response contrast (trigger vs generally unpleasant) would scale with misophonia severity across its entire spectrum.

## Materials and methods

### Participants

Thirty-five healthy participants (28 females; 18–52 y [median, 23 y]) participated in Experiment 1. Forty-four healthy participants (6 females; 18–64 y [median, 24 y]) participated in Experiment 2. All participants had normal or corrected-to-normal hearing and vision. Participants gave written informed consent and were remunerated by the hour or received credit points. Participants were recruited through University of Amsterdam student recruitment, (online) misophonia support groups, Dr. Rouw's misophonia research contacts, and word-of-mouth. The experiments were approved by the ethics committee of the Department of Psychology at the University of Amsterdam. Experiment 1 was approved on March 22nd, 2023 (FMG-2306) and ran from May 8th, 2023, until July 25th, 2023. Experiment 2 was approved on February 6th, 2025 (FMG-2306; amendment) and ran from February 20th, 2025, until May 16th, 2025. Five participants from Experiment 1 were excluded due to technical issues. Data from both experiments were pooled, unless stated otherwise.

### Procedure

Experiment 1 lasted for 1.5 hours and consisted of the following parts (explained in detail below) (Fig 1A): (i) misophonia protocol (block 1), (ii) breathwork/distraction intervention, (iii) misophonia protocol (block 2), (iv) frisson protocol (two blocks), and (v) post-experiment questionnaires. We collected eye and heart rate data during all sound presentations (steps i, iii and iv). Experiment 2 lasted for 1 hour and was identical to Experiment 1 except the breathwork intervention and frisson blocks were omitted. All acquired data is presented here, except for frisson data.

**Misophonia protocol.** All participants were asked to sit in a dimly lit room and to rest their head on a chin rest (placed at 50 cm distance in front of the screen) and to maintain central fixation. Each trial consisted of a sound interval, a rating interval and an inter-trial interval. The sound interval showed a blank grey screen and a fixation square, and a sound was presented (duration, 10–20 s; loudness, 30–70 dB). In the rating interval, the fixation square rotated 45° (into a diamond) and participants were prompted to rate how aversive the sound was, on a scale from 1 to 4, with a left-handed button press on the keyboard in front of them: G, "This sound doesn't affect me at all, the sound may be mildly annoying/disturbing"; H, "This sound is annoying, I don't like the sound"; J, "I do not like this sound at all, it makes me feel very uncomfortable"; Space bar, "I feel angry, I feel anxiety, I feel fear, I need to flee, I want to remove the headphones, this sound makes me physically upset". Participants received instructions about this mapping prior to starting the experiment. To avoid ambiguity the scale is ordered to reflect the increasing negative valence/increasing arousal characteristic to misophonia, from neutral to full misophonic response. The G-to-spacebar key arrangement was selected to support a comfortable hand posture and facilitate fast and intuitive response mapping. The trial ended with an inter-trial interval (2–3 s, uniformly distributed). Visual stimuli were displayed on a gamma-corrected monitor (spatial resolution of 2560 by 1440 pixels) with a vertical refresh rate of 100 Hz, and sound stimuli were presented using IMG Stageline MD-5000DR headphones.

In one block, participants were exposed to, and evaluated, ten generally unpleasant sounds, and ten common misophonia trigger sounds, which were each repeated once; the resulting forty sounds were presented in fully randomized order (Fig 1E). We used the following generally unpleasant sounds: baby crying, bicycle breaks, clarinet squeaks,

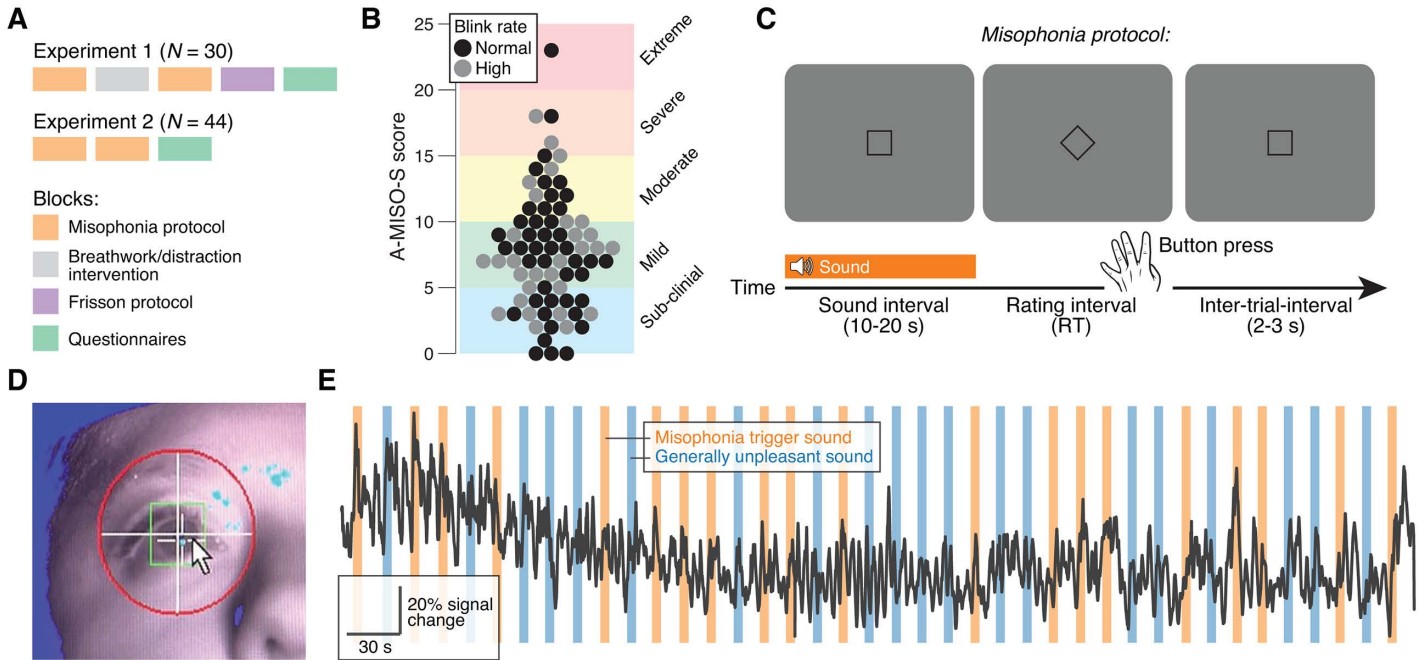

**Fig 1. Study design. (A)** Experimental blocks in experiments 1 and 2 (Materials and Methods). **(B)** Amsterdam Misophonia Scale (A-MISO-S) scores [3]. Every data point is a participant (N=74); grey data points indicate individuals who blinked on more than two-thirds of sound presentations (Materials and Methods). **(C)** Sequence of sound and rating interval during the misophonia protocol. **(D)** Image of pupil size recordings. **(E)** Example session. Black trace, pupil size time series; colored bars, sounds.

construction drilling, dog barking, fire alarm, glass on metal, modem dialup, nails on a chalkboard, and Styrofoam squeak. We used the following misophonia trigger sounds: apple eating, chip bag crinkling and eating (Cheetos), chip eating, clearing throat, dog eating, gum chewing, keyboard typing, pen clicking, runny nose, and slurping. All sounds were acquired copyright free from https://freesound.org/, except for the clarinet squeak which was recorded by K. Schwarz (one of the authors). We did not match sounds for all acoustic features, as this would distort sounds to the extent that they are no longer recognizable. For example, misophonia triggers are often soft human-made sounds (sniffing, swallowing and whispering), and loudness is an integral part of the sound stimulus. Therefore, we opted for using the same sounds as in previous work [1,41] and, for transparency, plotting the associated spectrograms (S1A Fig) and making the sound stimuli publicly available.

Generally unpleasant sounds occupy specific regions of acoustic space [41,42] (e.g., characteristic frequencies and amplitude modulations) that reliably evoke strong negative autonomic responses, making them robust control stimuli for the misophonic response. In contrast, misophonic trigger sounds are not defined by these "unpleasant" acoustic signatures: e.g., loudness is an aversive psychoacoustic property, but misophonic triggers are commonly soft [1–4,10]. Instead, the misophonic reaction critically depends on more complex cognitive, social, and neurological mechanisms [2,9,10,43], evoking a characteristic fast and high-intensity autonomic misophonic reaction that can feel almost uncontrollable. Consequently, the standard approach of contrasting these two sound categories [1,4,12,44–46] allows to measure the misophonic status of an individual's physiological and behavioral response to misophonic triggers over "normal" unpleasant sound responses (e.g., it controls for potential individual differences in hearing ability and general sound sensitivity).

**Breathwork/distraction intervention.** In Experiment 1, participants were randomly allocated into one of two experimental conditions: breathwork or distraction. Each intervention lasted 30 minutes. The breathwork intervention

was scripted and recorded by L. Alonso-Marmelstein (a certified breath coach and registered yoga teacher) and included multiple breathing techniques used in previous studies [47]. The specific breathing techniques were belly breathing, 4:4 breathing, 4:4:4:4 box breathing and 4:7:8 breathing. Each technique was presented for an average of 7.5 minutes. Participants were allowed to sit on a chair, cushion or lay down on a yoga mat for the duration of the intervention. We played the training video on a laptop that they could appropriately position. The distraction intervention consisted of continuously playing the game "Snake" from Google.com. The participant used the arrow keys to move a digital snake around the board, and the objective was to not crash into one's own tail, which kept growing. They were encouraged to beat the highest score set by other participants. The breathwork interventions did not affect any of the reported pupillometry results (S2L,M Fig).

**Frisson protocol.** In Experiment 1, participants were asked to provide a song that gave them chills, tingles down the spine, welling in the chest, or provided strong physiological sensations during listening. To control for effects of length, loudness, and musical features, the second song was the previous participant's chosen song [48]. Participants were asked to report the start (z button) and end (m button) of any physical feeling while listening. Frisson data were collected after the experimental conditions reported in this manuscript and do not relate to current results. Frisson data will be the focus of another report.

**Post-experiment questionnaires.** Participants filled out the Body Consciousness Scale [49] to measure awareness and monitoring of bodily processes, the Amsterdam Misophonia Scale [3] to measure misophonia severity, the STAI-S questionnaire to measure current state of mood [50] and a single question on the autonomous sensory meridian response [51]. While misophonia remains a topic of research, there is evidence that the current understanding and characterization of the condition can be reliably captured with the Amsterdam Misophonia Scale questionnaire [3,52,53] and that the scale is sensitive to variations in the severity of misophonic complaints, such as misophonic symptom reduction following treatment [54–56].

## Eye data acquisition

Pupil size and gaze data were obtained with Eyelink 1000 devices (SR Research, Osgoode, Ontario, Canada) at 1000 Hz with an average spatial resolution of 15–30 min arc (Fig 1D,E).

## Heart rate acquisition

Heart rate and heart rate variability were measured with an electrocardiogram (ECG) device. Three electrodes were placed on the participant's body: one on either side of their chest, and one on the lower left torso. Heart rate data will be the focus of another report.

## Exclusion criteria

We excluded trials in which participants blinked during the first three seconds of sound presentation and excluded participants for which more than two-thirds of trials had to be excluded due to blinking (N = 30, grey data points in Fig 1B and S1B,C Fig). The relatively high number of excluded participants may be explained by us not explicitly instructing them to refrain from blinking during the sounds. The average blink rate during the first three seconds of sound presentation was 19.5 blinks per minute (S1C Fig), which is close to the average human spontaneous blink rate of ~17 blinks per minute [57,58]. Results based on all data (without having excluded participants and trials) are presented in S2D,E Fig and warrant the same conclusions.

## Analysis of eye data

All analyses were performed using custom-made Python software.

**Blink and saccade detection.** Periods of blinks and saccades were detected using the manufacturer's standard algorithms with default settings. We counted the number of blinks that occurred during the first three seconds of each sound presentation.

**Preprocessing.** We applied to each pupil recording (i) linear interpolation of values measured just before and after each identified blink (interpolation time window, from 200 ms before until 200 ms after blink), (ii) band-pass filtering (third-order Butterworth, passband: 0.01–10 Hz), (iii) removal of pupil responses to blinks and to saccades, by first estimating these responses by means of deconvolution and then removing them from the pupil time series by means of multiple linear regression [59], and (iv) conversion to units of modulation (percent signal change) around the mean of the pupil time series from each block.

**Quantification of sound-evoked pupil responses.** We computed task-evoked pupil response measures for each trial as the mean of the pupil diameter modulation values in the window 1 s to 3 s from sound onset, minus the mean pupil size during the 0.25 s before sound onset. This time window is based on the known pupil impulse response function [60] and is the same as in recent pupillometry during sound presentation [61,62]. We additionally removed (via linear regression) trial-to-trial variation in the pupil response amplitude that was related to the pre-trial (pre-sound) baseline pupil size (e.g., through reversion to the mean [63]).

## Statistical comparisons

We used a paired samples t-test to quantify the difference in pupil response or subjective rating between trigger and generally unpleasant sounds. We used a mixed linear regression model to quantify the dependence of pupil response magnitude on aversiveness rating (values 1–4) and sound category (0, unpleasant; 1, trigger). The fixed effects were specified as:

$$P \sim \beta_0 1 + \beta_1 R + \beta_2 C$$

with $P$ as trial-wise pupil response magnitudes, $R$ as the sound-wise aversiveness ratings, $C$ as sound-wise sound category and $\beta$ as the regression coefficients. We included the maximal random effects structure justified by the design [64], which meant that intercepts and rating and category coefficients could vary with participant. We quantified across-participant correlations using Pearson's correlation coefficient.

## Results

### Misophonia severity is reliably predicted by trigger-evoked pupil dilation

We characterized misophonia severity in our sample of seventy-four participants (combined across experiments 1 and 2; Fig 1A) using the *Amsterdam Misophonia Scale* (A-MISO-S) [3] (Materials and Methods). In line with earlier work [9,16,20], we observed substantial variation in a sample drawn from the general population (Fig 1B): 27% was sub-clinical (scores 1–4), 43% experienced mild misophonia (scores 5–9), 21% experienced moderate misophonia (scores 10–14), 7% experienced severe misophonia (scores 15–19) and 1% experienced extreme misophonia (scores 20–24). The same seventy-four participants listened to a sequence of sounds and rated their aversiveness on a four-point scale (Fig 1C; Materials and Methods): half were generally unpleasant (e.g., nails on chalkboard) and half were common misophonia triggers (e.g., clearing throat). See Materials and Methods for full list of sounds and S1A Fig for their spectrograms. We simultaneously recorded pupil size (Fig 1D,E).

We observed robust pupil dilation in response to both misophonia trigger and generally unpleasant sounds (Fig 2A). On average, sound-evoked pupil dilation was larger for misophonia trigger versus generally unpleasant sounds (Fig 2A,B). Given the more subtle acoustic features of misophonia trigger sounds (S1A Fig; Materials and Methods), this group-level

 

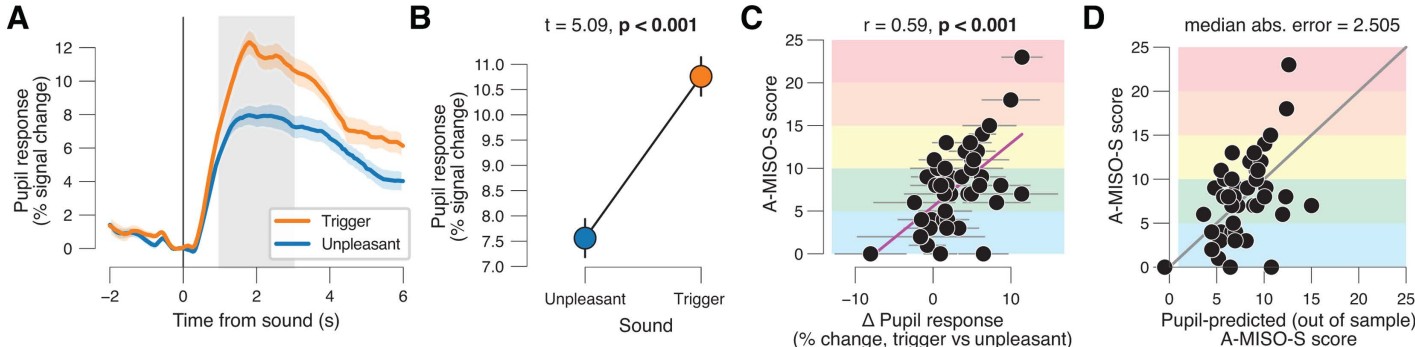

**Fig 2. Sound-evoked pupil dilation reliably predicts individual misophonia severity. (A)** Pupil response time courses separately for sound category (misophonia trigger and generally unpleasant; collapsed across rating), time-locked to sound onset. Shading, S.E.M. across participants (N = 44); grey shading, interval used for quantifying task-evoked pupil responses (Materials and Methods) **(B)** Sound-evoked pupil response magnitude, separately sound category. Error bars, S.E.M. across participants (N = 44); stats, paired samples t-test. **(C)** Individual A-MISO-S score plotted against individual pupil response contrast (trigger vs unpleasant sounds). Each data point is a participant. Error bars, standard deviation of bootstrapped distribution, corresponding to S.E.M; stats, Pearson correlation. **(D)** A-MISO-S score plotted against pupil-predicted A-MISO-S score (leave-one-out cross-validation; Materials and Methods). Each data point is a participant. Solid line, identity line.

effect may seem unexpected at first glance. However, this effect may be explained by the substantial number of partici-pants in our sample with mild to severe misophonia (Fig 1B; average A-MISO-S score of our sample is 8 out of 25).

We next tested if, across individuals, these sound-evoked pupil responses were related to misophonia severity. In line with earlier work [1,4,12,44–46], we controlled for potential confounding individual differences such as hearing abil-ity or general sound sensitivity by computing a pupil response contrast (Materials and Methods): misophonia trigger vs generally unpleasant. This contrast is a standard method in misophonia research, as it specifically reflects the effect of the misophonic trigger, rather than more general emotional, auditory or physiological (pupil) effects. As a first step, we assessed test–retest reliability of pupil responses separately for triggers and generally unpleasant sounds, and for the contrast between them. Reliability was high for trigger sounds (r = 0.805) and generally unpleasant sounds (r = 0.758) but lower for the contrast (r = 0.570) (S2A Fig). This reduction in reliability is consistent with the well-known tendency for difference scores to accumulate measurement error from both component conditions. Next, confirming our hypothesis, we found that the pupil response contrast reliably predicted A-MISO-S scores across participants (Fig 2C). We used leave-one-out cross-validation to quantify the error between pupil-predicted misophonia severity and actual misophonia severity. Specifically, we computed the linear relationship between the A-MISO-S scores and the pupil response contrast (trigger vs unpleasant) based on the data from all participants minus one and then predicted the A-MISO-S score for the left-out participant based on his/her pupil data. This analysis showed that A-MISO-S score could be reliably predicted, based on only pupil dilation, at the level of a single individual (Fig 2D): the median absolute error was 2.505, which is below the granularity of severeness categories of the A-MISO-S.

Several control analyses speak to the robustness of our results. First, we provided an internal replication. In all analy-ses reported so far, we pooled data from Experiments 1 and 2 (Fig 1A; Materials and Methods), but, critically, A-MISO-S scores and the pupil response contrast are robustly correlated in two independent groups of participants (S2B,C Fig). Second, the result is robust with respect to how eye blinks are treated in the analyses. In all analyses so far, we excluded trials in which participants blinked during the first three seconds of sound presentation (Materials and Methods) and excluded participants for which more than two-thirds of trials had to be excluded due to blinking (N = 30, grey data points in Fig 1B and S1B,C Fig; Materials and Methods), but the results are qualitatively the same when including all participants and trials regardless of blink behavior (S2D,E Fig). Third, the result is robust with respect to pupil data preprocessing

choices. We preprocessed the pupil data in the same way as in our recent pupillometry work [65–67] (Materials and Methods), but the results are qualitatively the same when doing less extensive preprocessing (S2F,G Fig). Fourth, using 0–10 seconds as a broader time window of interest to compute sound-evoked pupil responses resulted in an even stronger correlation between A-MISO-S scores the pupil contrast (r = 0.64, p < 0.001). This result may indicate that sound stimuli evoke alertness maintenance over relatively long timescales [68]. Fifth, the result cannot be explained by differences in (micro)-saccade rates (S2H,I Fig).

One possibility is that pupil response variability is smaller for trigger vs generally unpleasant sounds, and that this is especially true for those individuals with severe misophonia. However, we did not find evidence for this: at the group-level there was no difference in pupil response variability between triggers and generally unpleasant sounds (S2J Fig); likewise, there was no significant relationship between A-MISO-S scores and individual pupil response variability contrast (S2K Fig).

In sum, we confirmed our hypothesis that misophonia severity is predicted by evoked pupil dilation to triggers vs generally unpleasant sounds. This suggests that a single individual with unrealistic questionnaire scores (e.g., due to misinterpretation or response bias) might be detectable by adding pupillometry as an objective tool.

### Trigger-evoked pupil dilation and subjective aversiveness ratings each explain unique variance in misophonia severity

After each sound presentation, participants rated its aversiveness on a four-point ordinal scale (Fig 1C), ranging from neutral to full misophonic response (Fig 3A; Materials and Methods). Because of the chosen definitions associated with each rating (Materials and Methods), we assumed that they are ordered with interval properties. This has the benefit that we could repeat the same analyses as we performed for sound-evoked pupil responses.

Three analyses indicated that subjective ratings and evoked pupil responses are related. First, we observed larger pupil responses evoked by sounds that were rated as more aversive (S3A Fig). Second, mimicking the group-level pupil responses (Fig 2A, B), subjective ratings were higher (more negative) for misophonia trigger versus generally unpleasant sounds (Fig 3B). Third, again mimicking the pupil results (Fig 2C), A-MISO-S scores were positively related to the subjective rating contrast (Fig 3C).

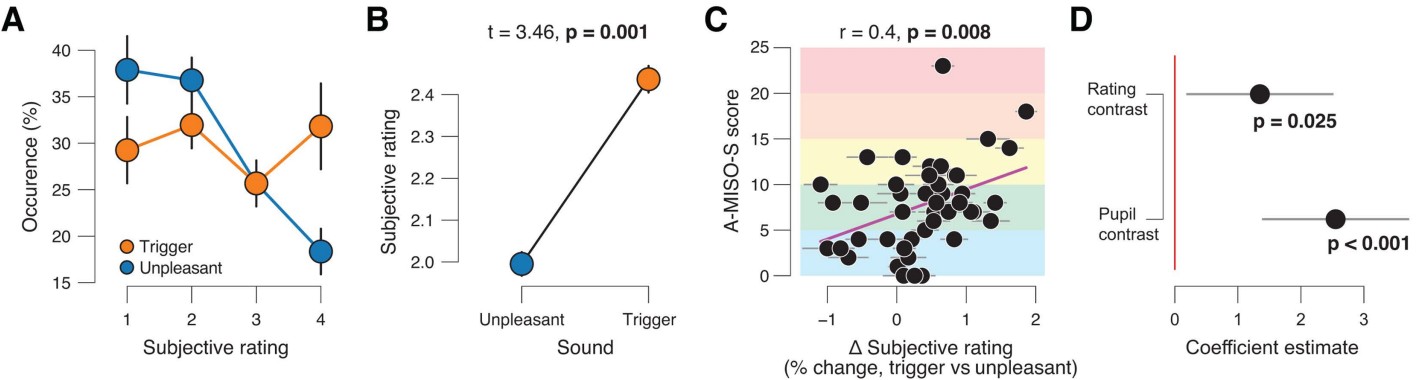

**Fig 3. Sound-evoked pupil dilation reliably predicts individual misophonia severity. (A)** Subjective rating occurrence, separately for misophonic triggers and generally unpleasant sounds. A higher number reflects a more aversive subjective rating (Materials and Methods). Error bars, S.E.M. across participants (N = 44). **(B)** Mean subjective rating, separately for misophonic triggers and generally unpleasant sounds. Error bars, S.E.M. across participants (N = 44); stats, paired samples t-test. **(C)** Individual A-MISO-S score plotted against individual subjective rating contrast (trigger vs unpleasant sounds). Each data point is a participant. Error bars, standard deviation of bootstrapped distribution, corresponding to S.E.M; stats, Pearson correlation. **(D)** Coefficients of across-subjects multiple regression model in which we regressed A-MISO-S scores on pupil response and subjective rating contrasts (both trigger vs generally unpleasant). Error bars, 95% confidence interval.

However, two results show that aversiveness ratings and pupil responses can also diverge. First, sound-evoked pupil responses were also larger for misophonia trigger sounds when matched for subjective aversiveness rating (S3B Fig; mixed linear model: main effects for aversiveness rating [$z=6.790$, $p<0.001$] and sound category [$z=3.031$, $p=0.002$]). Thus, for the same subjective aversiveness rating, the pupil dilated more in response to misophonia trigger sounds, than generally unpleasant sounds. In this regression model the average (± S.E.M.; across participants) variance inflation factor (VIF) is 1.27 (± 0.06), which is considerably lower than common cutoffs 5 or 10. Second, the test-retest reliability of subjective ratings was higher than that of pupil responses (trigger sounds, $r=0.942$; generally unpleasant sounds, $r=0.920$; contrast, $r=0.899$) (S3C Fig). Third, we observed that the pupil response contrast (trigger vs unpleasant) was not significantly correlated to the aversiveness rating contrast (trigger vs unpleasant) across participants (S3D Fig). In other words, those participants who rated the misophonia trigger sounds as more aversive than the generally unpleasant sounds did not necessarily also have larger pupil responses during trigger versus unpleasant sounds. Thus, evoked pupil responses and subjective ratings could, in principle, predict unique parts of the across-subjects variance in misophonia severity, as measured by the A-MISO-S.

We used multiple linear regression to assess the unique contribution of subjective ratings and pupil responses in predicting A-MISO-S scores. In this multiple regression model, both measures significantly predicted misophonia severity (Fig 3D; pupil response contrast, $t=4.414$, $p<0.001$; aversiveness ratings contrast, $t=2.335$, $p=0.025$). We observed the same qualitative result when excluding one participant with severe misophonia (A-MISO-S score > 20) from the analysis, who could be considered an outlier in our sample (pupil response contrast, $t=3.580$, $p=0.001$; aversiveness rating contrast, $t=2.542$, $p=0.015$).

In sum, sound-evoked pupil dilation and subjective aversiveness ratings each explain unique variance in misophonia severity. This highlights the importance of adding an objective physiological measurement to misophonia diagnosis.

## Discussion

Our study demonstrates for the first time that pupillometry is a sensitive method for measuring variations in misophonic responses, even in milder cases. Pupil response discerned misophonic responses to "trigger" sounds from normal aversive responses to generally unpleasant sounds. Furthermore, pupillometry proved sensitive to variations in the intensity of the misophonic response across different individuals and may therefore be used to aid diagnosis of the condition.

Our finding that pupillometry can reliably measure (variations in) misophonic severity suggests it is a valuable addition to self-report paradigms. While self-report is an important tool to measure phenomenological and emotional experiences, it has some intrinsic weaknesses including unintentional (e.g., misunderstandings or biases) and intentional (malingering) measurement errors. Furthermore, our results show how the (continuous) pupillometry variance can explain additional variance over the (categorical) ratings for misophonic sounds. Thus, adding pupillometry to the misophonia diagnostic toolbox could greatly help deciphering out the different types and severities within a group of individuals reporting misophonic complaints.

Previous studies measured increased autonomic nervous system response in galvanic skin response and heart rate variability [1,4,11,39] and shown how the involuntary emotional and physiological response is an essential characteristic of misophonia [10]. In these previous studies, only the autonomic nervous system response to misophonic sounds in misophonic versus non-misophonic participants was obtained. Our current goal was to add an objective measurement that would provide a more sensitive measurement of across-subject variations in autonomic nervous system response. Pupil size fluctuations at constant luminance have previously been shown to reflect neuromodulatory activity and emotional arousal associated with increased sympathetic activity [28,31–34]. The signal-to-noise ratio of pupil responses is higher compared to galvanic skin heart rate (variability) responses [40]. Furthermore, the temporal resolution of pupillometry is higher compared to functional MRI and physiological measures like the galvanic skin response. Together, this allowed us to take previous findings a step further and predict that pupillometry measurements reflect variations in the strength

of misophonic responses (between participants, or between trials). Indeed, our results reflected such response variation, even in moderate and mild cases of misophonia.

There is also a practical reason to add pupillometry to the misophonia toolbox. As compared with neuroimaging or electrophysiological techniques, pupillometry is cheap, mobile, and non-invasive. This is not only relevant to misophonia research, as a similar quest for objective measurements exist in the pluriform and extensive research on sound sensitivity in related research areas (e.g., autism, tinnitus, hyperacusis, or PTSD [69–71]. A recent preprint showed that emotionally evocative sounds elicited abnormally large pupil dilations in a "disordered hearing" (tinnitus and hyperacusis) group [72]. This invites further research of employing pupillometry in sound sensitivity in other conditions. We propose that pupillometry could be employed as a non-invasive, relatively easy-to-use and cost-effective manner to map out and compare responses both within and across the different conditions. An additional advantage is that this methodology can even be used in cases where self-report is limited (e.g., due to age effects or language impairments of the participant group).

There are, however, several limitations to the use of pupillometry in this context. First, there are known time-scale effects on the pupil size timeseries [68,73,74]. However, since we fully randomized sounds within blocks, time-scale effects could have only added noise and could not have driven the result. Second, the signal-to-noise ratio of pupil responses is limited, as illustrated by the modest test-retest reliability of the pupil contrast (S2A Fig; ICC = 0.497). This level of reliability implies that the observed correlation with misophonia severity is likely an underestimate of the true association, because measurement error in the predictor attenuates observed correlations. Thus, the reliability estimate qualifies the precision of the effect size while supporting, rather than undermining, the robustness of the reported relationship. Third, pupillometry is relatively unspecific. For example, pupil dilation can also reflect positive emotional reactions to sounds in frisson [48] and the autonomous sensory meridian response [51]. We dealt with this by presenting sounds that would only evoke negative responses (Materials and Methods), and by computing the sound-evoked pupil contrast (trigger vs generally unpleasant) [1,4,12,44–46], as it controls for potential individual differences such as hearing ability and general sound sensitivity. At the level of the central nervous system, it has been shown that pupil size fluctuations at constant luminance reflect the activity of the cortical salience network (including the insula) [28] as well as that of a number of subcortical neuromodulatory structures [34], including the noradrenergic locus coeruleus [28,35,75,76], the cholinergic basal forebrain [28,63,75], and likely others [77,78]. Thus, future studies should pinpoint the exact neuroanatomical and neurochemical source(s) of our observed effects. Similarly, the current study did not examine which cognitive, social, or emotional factors are associated with the increased pupil response to misophonic sounds (e.g., disgust [38,79]). Exploring the role of such factors with pupillometry represent important directions for future misophonia research.

There are several further limitations to our study. First, follow-up studies can elucidate whether choices in experimental design may increase or decrease obtained effects. In particular, a different set of misophonic/unpleasant sound stimuli may affect the magnitude of the autonomic nervous system (and therefore pupil) response. For example, we choose to use the same standard stimuli for all participants, but using participants' individual misophonic trigger sounds may lead to different results. However, we expect that such a change in design would only enhance our current results. Second, differences in low-level acoustic features may have affected our group-level analyses. Matching for all acoustic features is however impossible, as this would distort sounds to the extent that they are no longer recognizable (Materials and Methods). Critically, our main conclusions rely on individual differences analyses; since all participants were exposed to the exact same sounds, these analyses do not suffer from potential systematic differences in low-level acoustic features. Third, analyzing which trigger sound most strongly drives pupil responses in an individual, or a more in-depth analysis of individual differences in misophonia was not feasible in the current design, due to limitations in the number of trials and participants. This remains a topic for future research. Fourth, the subjective ratings were measured on a four-point ordinal scale, while evoked-pupil responses were measured on a continuous scale, which may have favored the pupil predictor in the multiple linear regression model. Future work could measure subjective ratings on a continuous scale. Finally,

because of the recruitment method and limited number of participants, our results cannot be used to estimate misophonia prevalence in the general population.

The current results reflect how even milder misophonic cases can experience emotional disturbances related to "triggers" in their normal daily life. There are practical implications to these findings, as for sufferers from misophonia the relatively unknown and ill-understood status of the condition impedes explaining misophonia to others, or even understanding their misophonic responses themselves [1,3,7,9]. Adding pupillometry as an objective misophonia measurement tool can improve this situation. Pupil dilation can clearly display the physical response of the sufferers to the triggers. Such objective validation can bring relief to sufferers and improve understanding from their environment. Furthermore, the current lack of objective diagnostic tools hinders rigorous assessment of the efficacy of different treatment approaches of misophonia [8,10]. Our findings suggest that pupillometry can help assess strategies and evaluate new treatments and thus help build effective treatments for this debilitating condition.

## Supporting information

**S1 Fig. (A) Spectrograms of all sounds.** (B) Blink rate during first three seconds of sound presentation. Data is from all participants and all trials (N = 74; before exclusion; Materials and Methods). (C) Histogram of percentage of trials in which participants blinked during the first three seconds of sound presentation (Materials and Methods). Red dashed line, cut off for participant exclusion (66.7%; Materials and Methods).
(EPS)

**S2 Fig. (A) Bootstrapped distribution of test-retest reliability of pupil measures.** Colored tick marks, median of distribution. We bootstrapped the data 10K times separately per participant (with replacement; until the same data set size) and computed the average participant-wise pupil response evoked by triggers, evoked by generally unpleasant sounds, and their contrast. We then correlated pupil measures of neighboring bootstraps across participants and build a bootstrapped distribution of correlation coefficients. Additionally, we computed the intra-class correlation coefficient [80] of the pupil contrast between odd and even sound presentations, based on a mean-rating (k = 2), absolute-agreement, 2-way mixed-effects model: ICC = 0.497, F(43,43) = 1.920, p = 0.018, CI95% = [0.05,0.72]. (B,C) As main Fig 2C, but separately for Experiments 1 and 2 (Materials and Methods). (D,E) As main Fig 2A,C but for all data from all participants and all trials (N = 74; before exclusion; Materials and Methods). (F,G) As main Fig 2A,C but after less extensive preprocessing of pupil data: only blink interpolation and conversion to percent signal change. (H,I) As main Fig 2B,C but for saccade rate during the first three seconds of sounds presentation. (J,K) As main Fig 2B,C but for pupil response variability (s.d.). (L) Pupil response (collapsed across sound category) in Experiment 1, separately per experimental manipulation (breathwork [N = 14] vs distraction [N = 8]) and block number. Breathwork reduced the magnitude of sound-evoked pupil responses (irrespective of sound category) in block 2 vs block 1 (p = 0.006). Distraction had no such effect (p = 0.621). The breathwork-related change in pupil response magnitude was not larger than the distraction-related change (p = 0.821). (M) Pupil response contrast (trigger vs unpleasant sounds) in Experiment 1, separately per experimental manipulation (distraction versus breathwork) and block number. There was no effect of breathwork (p = 0.096) or distraction (p = 0.687) on pupil response contrast in block 2 vs block 1, and no difference in this effect between conditions (p = 0.353).
(EPS)

**S3 Fig. (A) Pupil response time courses separately for subjective rating (of physical and emotional discomfort on a 4-point scale; collapsed across sound category), time-locked to sound onset.** Grey shading, interval used for quantifying task-evoked pupil responses (Materials and Methods). (B) Sound-evoked pupil response magnitude, separately per rating and sound category. (C) Bootstrapped distribution of test-retest reliability of rating contrast (Methods). Colored tick marks, median of distribution. Additionally, we computed the intra-class correlation coefficient [80] of the pupil contrast between odd and even sound presentations, based on a mean-rating (k = 2), absolute-agreement, 2-way

mixed-effects model: ICC = 0.869, F(43,43)=1.920, p<0.001, CI95%=[0.76,0.93]. (D) Individual pupil response contrast (trigger vs unpleasant sounds) plotted against individual subjective rating contrast (trigger vs unpleasant sounds). Each data point is a participant. Error bars, standard deviation of bootstrapped distribution, corresponding to S.E.M. Stats, Pearson correlation.

(EPS)

## Acknowledgments

We are grateful to all participants. We thank Chelsey Kret and Valentina Zhang for help with data collection. We thank E.H.F de Haan, K.R. Ridderinkhof and two reviewers for their insightful comments which improved the manuscript.

## Author contributions

**Conceptualization:** Jan Willem de Gee, Romke Rouw.

**Formal analysis:** Jan Willem de Gee.

**Investigation:** Laia Alonso-Marmelstein, Kate Schwarz-Roman.

**Supervision:** Jan Willem de Gee, Romke Rouw.

**Writing – original draft:** Jan Willem de Gee, Romke Rouw.

**Writing – review & editing:** Jan Willem de Gee, Laia Alonso-Marmelstein, Kate Schwarz-Roman, Romke Rouw.

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
