## [Decision Letter · Decision Letter 0]

9 Sep 2025

PONE-D-25-31481Sound-evoked pupil dilation quantifies misophonic symptomsPLOS ONE

Dear Dr. de Gee,

Thank you for submitting your manuscript to PLOS ONE. After careful consideration, we feel that it has merit but does not fully meet PLOS ONE’s publication criteria as it currently stands. Therefore, we invite you to submit a revised version of the manuscript that addresses the points raised during the review process. Background: I was fortunate to obtain two excellent expert reviews and have also read the manuscript myself. Both reviewers find the study promising in terms of its general idea and consider the use of pupil dilation as a potential objective measure of misophonic responses to be of future value, and I fully agree. However, both raise at the same time several important issues that should be addressed in a revision. Most importantly, the theoretical basis needs to be elaborated in more details: what mechanism is assumed to underlie the observed pupil size effects, why these should differ between trigger and unpleasant sounds, and what this implies conceptually. Closely related is the question of what exactly the rating scales measure, what the categories mean, and how they should be interpreted analytically. In addition, both reviewers provide a list of methodological points, which I do not reiterate here. Overall, I ask you to prepare a revised manuscript together with a point-by-point response addressing all reviewer comments.

 Please submit your revised manuscript by Oct 24 2025 11:59PM. If you will need more time than this to complete your revisions, please reply to this message or contact the journal office at plosone@plos.org. Please include the following items when submitting your revised manuscript:

We look forward to receiving your revised manuscript.

Kind regards,

Michael B. Steinborn, PhD

Section Editor

PLOS ONE

Journal Requirements:

J.W. de Gee is supported by the NWO Talent Programme which is (partly) financed by the Dutch Research Council (NWO) under the grant VI.Veni.232.210.

3. Please expand the acronym “NWO” (as indicated in your financial disclosure) so that it states the name of your funders in full.

5. We notice that your supplementary figures are included in the manuscript file. Please remove them and upload them with the file type 'Supporting Information'. Please ensure that each Supporting Information file has a legend listed in the manuscript after the references list.

Reviewers' comments:

Reviewer's Responses to Questions

**Comments to the Author**

1. Is the manuscript technically sound, and do the data support the conclusions?

Reviewer #1: Partly

Reviewer #2: No

2. Has the statistical analysis been performed appropriately and rigorously? 

Reviewer #1: Yes

Reviewer #2: Yes

3. Have the authors made all data underlying the findings in their manuscript fully available?

Reviewer #1: No

Reviewer #2: Yes

4. Is the manuscript presented in an intelligible fashion and written in standard English?

Reviewer #1: Yes

Reviewer #2: Yes

5. Review Comments to the Author

Reviewer #1: Title: Sound-evoked pupil dilation quantifies misophonic symptoms

Authors: Jan Willem de Gee, Laia Alonso-Marmelstein, Kate Schwarz-Roman, Romke Rouw

Journal: PLOS One

Reviewer: Jens Kürten

Background

The study presented here investigates the potential of pupillometry as an objective physiological marker for misophonia, both in terms of distinguishing it from general sound unpleasantness and in predicting the severity of misophonic symptoms within individuals. The authors define misophonia as a condition characterized by intense negative emotional, cognitive, or physiological responses to common, human-made sounds such as chewing or throat clearing.

In two experiments, participants listened to sounds (presumably presented in randomized order) categorized by the authors (a priori) as either “generally unpleasant” or “misophonia triggers”, while their pupil size was continuously recorded using high-temporal-resolution eye-tracking. After each sound, participants rated its unpleasantness on a four-point categorical scale with the options “neutral” (1), “mild dislike” (2), “disgust or anger” (3), and “full misophonic reaction” (4). The authors report the following key findings: First, misophonic trigger sounds were rated as more unpleasant than generally unpleasant sounds. Second, pupil dilation from sound onset was greater for misophonic trigger sounds than for generally unpleasant sounds, even when matched on subjective aversiveness ratings. Finally, individual differences in both pupil dilation and rating contrasts predicted participants’ scores on the Amsterdam Misophonia Scale (A-MISO-S), suggesting that both measures may have diagnostic utility for assessing symptom severity.

Evaluation

My overall impression of the manuscript is mixed. The topic is timely and relevant, and the manuscript is generally well written and clearly structured. The statistical analyses appear rigorous, and the visualizations are informative and well-designed. However, there are several conceptual and methodological issues that should be addressed before I can recommend the manuscript for publication. My main concerns pertain to the theoretical justification for the pupillometry approach and the key comparison, the need for clarification of specific methodological procedures, and questions regarding the interpretation of the physiological data.

Major points

Theoretical motivation

The authors motivate their work by emphasizing the need for objective indicators of misophonia due to its high prevalence and clinical relevance. However, the study would benefit from a stronger theoretical justification for using pupil dilation (rather than alternative physiological measures) as an indicator of the condition, and for the specific comparison between misophonic and generally unpleasant sounds. Pupil dilation is commonly interpreted as a marker of arousal or cognitive effort, but these are broad and relatively unspecific constructs. As the authors themselves acknowledge, pupil responses can be driven by a wide range of processes, including positive emotional reactions, not just negative responses such as those associated with misophonia.

Furthermore, it was unclear from the introduction whether the authors expected both higher aversiveness ratings and greater pupil dilation for misophonic trigger sounds compared to generally unpleasant sounds, as the directional hypotheses were not explicitly stated. Similarly, it is not explained why pupil responses should scale with the severity of misophonic symptoms. The manuscript would benefit from a clearer theoretical framework outlining what pupil size is thought to index in this context, and how the mechanisms underlying misophonic versus generally aversive sound processing differ. Integrating these theoretical perspectives more directly into the introduction could provide a stronger rationale for the current research and the critical comparisons.

Methodology and results

Regarding methodology and the interpretation of the pupil data, my first concern relates to the control of low-level stimulus properties. From my understanding (though I may be mistaken) the sounds differed within and across categories in terms of duration (10–20 seconds), loudness (30–70 dB), and likely in additional acoustic features such as onset regularity or sharpness (based on the spectrograms provided). How can it be ruled out that differences in aversiveness ratings and pupil dilation were at least partly driven by systematic differences in these low-level features? Did the authors take any steps to control such confounds? In addition to the visual representations provided, I would strongly encourage the authors to make the sound stimuli and data available (at least to reviewers) to allow for better evaluation.

I also have some concerns about the subjective aversiveness ratings. The four response options appear to be more qualitative than ordinal, which complicates the interpretation of mean scores (e.g., as shown in Figure 2A). Why did the authors not opt for a more conventional Likert-type scale ranging from, for instance, 1 to 5, with clearly ordered categories for aversiveness or valence, or a continuous visual analogue scale? Furthermore, it is unclear why the lowest anchor was “neutral”. Is it possible that some participants found individual sounds pleasant but had no way to indicate this due to the restricted response range?

Participants were reportedly instructed on how to use the scale, including what constitutes a “full misophonic reaction”. This definition should be included in the manuscript for readers unfamiliar with the concept of misophonia. More critically, the explicit inclusion of a “full misophonic reaction” category, especially since it was mapped to a distinct response key, may have introduced demand characteristics or bias. Could this have turned the task into something akin to a “misophonic trigger detection” task, potentially influencing both behavioral and physiological responses? For instance, did participants potentially expend more cognitive effort to determine whether a sound warranted the “full misophonic reaction” label? Additionally, given the instructions regarding the construct and if the generally unpleasant sounds did not contain any misophonia triggers, one would not expect any generally unpleasant sounds to receive a rating of 4. Was this the case?

Given the categorical and arguably non-metric nature of the scale, I would also strongly encourage the authors to show the full distribution of ratings by sound category rather than presenting mean values, which may be inappropriate for the level of measurement.

Minor points

1. Section 2: The study appears to be embedded within a larger research program, given the variety of manipulations, protocols, and measures mentioned but not analyzed in the current paper. While this is not problematic per se, I would expect a preregistration or study plan indicating that the current subset of analyses and measures was part of the original plan. Without such documentation, the selection of outcomes may appear post hoc. To be clear, I do not intend to imply selective reporting, indeed, the authors are very transparent about their design choices and analyses, but it may be helpful to clarify this point earlier in the manuscript. I would prefer the full data from all protocols and conditions to be made available to the reviewers before publication.

2. Section 2.2.1 / Figure 1: The study used 10 different “generally unpleasant” and 10 different “misophonic trigger” sounds. From the description of the misophonia protocol, I gather that the two misophonia blocks were separated by a breathwork/distraction intervention in Experiment 1. Figure 1D suggests that the sounds were randomly intermixed within blocks, with 20 sounds per block. Did each sound appear once per block, or twice? This could be clarified in the protocol description, and the figure could indicate block structure more clearly.

3. Section 2.3: Were participants given fixation instructions? Since pupil size is sensitive to gaze direction, it would be important to report whether participants were instructed to maintain central fixation and whether gaze position was monitored during the trials.

4. Section 2.5.2: It was unclear why trials with blinks during the first three seconds were excluded if pupil size during blinks was interpolated. Could the authors clarify this criterion?

5. Section 2.7: The authors used linear mixed-effects models to predict pupil dilation based on sound category and average aversiveness ratings. In addition to the general concerns raised earlier about the rating scale, I am also wondering about potential collinearity between predictors, given that ratings and sound categories were likely correlated. Was this considered or tested?

Reviewer #2: The authors investigate misophonia, a disorder where certain sounds evoke intense negative arousal responses. The study employs pupillometry to objectively measure these responses, which show distinctive pupil dilation patterns. This method could differentiate misophonic responses from other unpleasant sounds and reflect the severity of symptoms in individuals.

This manuscript presents an interesting and novel approach to objectively assess misophonia through pupillometry. However, several key concerns and clarifications are needed before the manuscript can be considered for publication. While the study design and methodology are promising, the authors should refine their explanations and analyses to ensure that the results are fully interpretable and robust. Also, the manuscript would benefit from a more thorough discussion of the limitations and potential sources of bias in the data.

1. Data Preprocessing: The extensive preprocessing applied to the pupil diameter data raises some concerns. While rigorous noise removal is important, overly complex preprocessing could reduce the transparency and reproducibility of the analysis. Would similar results be obtained with a simpler preprocessing approach, such as blink removal and linear interpolation, applied to the pupil data?

2. Potential Confounding Factors: The study could benefit from addressing the potential impact of time-scale effects on pupil dilation. One potentially uncontrolled factor is the time-scale effects on pupil diameter. It is known that arousal fluctuations influencing task performance can change with rhythms in the 10-16 second range (https://doi.org/10.1523/ENEURO.0250-24.2024), and the stimuli in this study were presented in cycles that align with this period. Additionally, reports suggest that pupil diameter tends to decrease over the course of a long experiment (https://doi.org/10.1371/journal.pone.0165274), as also seen in Figure 1D. These points should at least be mentioned as a limitation in the study.

3. Possible Setup issue?: The exclusion of about 30 participants due to blinks is notable. While it is good that the authors have shown that the results remain the same without these exclusions, why were such a large number of blink intervals recorded for most participants? It seems unusual for participants to blink this often everyday activities. Is there something about the (unusual) experiment setup that could be contributing to this?

4. Stimulus Variability: The current analysis primarily focuses on individual differences. It would be beneficial to also examine intra-individual variability in response to different stimuli. The authors mention that the sample size is too small to examine differences between stimuli, but can any statistical measures of variability be computed between stimuli? Even without a full stimulus-by-stimulus analysis, could provide valuable insights into misophonic responses.

5. Rating Methodology: It seems that four distinct qualitative categories were converted into a quantitative scale (1-4). The authors should justify this methodology or consider alternative ways of analyzing subjective ratings of stimuli.

6. Time Window for Pupil Dilation Measurement: The rationale for the specific 1-3 second window following stimulus presentation should be explained in more detail. A clearer justification would help readers understand the significance of this timing.

7. Operational Definitions: A key issue I had concerns the operational definition of what distinguishes “unpleasant” sounds from “trigger” sounds. The operational definitions of these categories should be well-defined and justified to avoid ambiguity in the results. If the A-MISO-S score is treated as a reference for "true" misophonic severity, is this assumption valid? Is the difference between "trigger" and "unpleasant" sounds based on a specific, clear criterion, and how does this relate to the results?

8. Sound-evoked Pupil Responses: Even if there were clear, operational criteria for distinguishing unpleasant from trigger, it remains questionable whether the contrast between unpleasant and trigger itself produced this difference. It is unclear whether other factors, such as loudness, subjective saliency (https://doi.org/10.3758/s13423-015-0898-0) or emotional responses (https://doi.org/10.1016/j.biopsycho.2025.109044), could be contributing to these effects. After controlling all these factors, is it valid to conclude that the contrast between unpleasant and trigger explains the observed differences in pupil dilation?

9. Theoretical Considerations: Lines 323-324 suggest that the combination of a categorical scale with continuous pupil diameter data improves explanatory power. However, isn't it possible that these measures are assessing different aspects? The relationship between pupil diameter and arousal, as well as the differences between continuous and categorical measures, could be more thoroughly explored in the discussion. The theoretical basis for combining these measures should be better explained.

Minor Comments:

Lines 251-252: Lines 251-252 are unclear. Could the authors clarify what sample is referred to as 8 and what the sentence is meant to convey?

6. PLOS authors have the option to publish the peer review history of their article (what does this mean?). If published, this will include your full peer review and any attached files.

Reviewer #1: **Yes:**Jens Kürten

Reviewer #2: No

---

## [Author Response · Author response to Decision Letter 1]

12 Jan 2026

See cover letter and response to reviewers files

---

## [Decision Letter · Decision Letter 1]

4 Mar 2026

PONE-D-25-31481R1Sound-evoked pupil dilation quantifies misophonic symptomsPLOS One

Dear Dr. de Gee,

Thank you for submitting your manuscript to PLOS ONE. After careful consideration, we feel that it has merit but does not fully meet PLOS ONE’s publication criteria as it currently stands. Therefore, we invite you to submit a revised version of the manuscript that addresses the points raised during the review process.

We look forward to receiving your revised manuscript.

Kind regards,

Michael B. Steinborn, PhD

Section Editor

PLOS One

Journal Requirements:

Reviewers' comments:

Reviewer's Responses to Questions

**Comments to the Author**

1. If the authors have adequately addressed your comments raised in a previous round of review and you feel that this manuscript is now acceptable for publication, you may indicate that here to bypass the “Comments to the Author” section, enter your conflict of interest statement in the “Confidential to Editor” section, and submit your "Accept" recommendation.

Reviewer #1: All comments have been addressed

Reviewer #2: All comments have been addressed

2. Is the manuscript technically sound, and do the data support the conclusions?

Reviewer #1: Yes

Reviewer #2: Yes

3. Has the statistical analysis been performed appropriately and rigorously? 

Reviewer #1: Yes

Reviewer #2: Yes

4. Have the authors made all data underlying the findings in their manuscript fully available?

Reviewer #1: Yes

Reviewer #2: Yes

5. Is the manuscript presented in an intelligible fashion and written in standard English?

Reviewer #1: Yes

Reviewer #2: Yes

6. Review Comments to the Author

Reviewer #1: I have reviewed a previous version of this manuscript and appreciate the substantial effort in this revision, particularly in clarifying the methods and demonstrating the robustness of the findings. Most of my prior concerns have been adequately addressed, though a few points warrant further clarification to strengthen the paper prior to publication.

My primary remaining question concerns the pupil size contrast between misophonic triggers and generally aversive sounds. I now better understand the focus on individual differences in these contrasts rather than the group-level effect per se. That said, it was surprising to observe such robust correlations between the difference scores, which are typically less reliable than raw scores, and the rating scores. Could the authors provide reliability estimates for the pupil contrast (e.g., split-half reliability)? More importantly, however, i think that discussing potential reasons for the observed pupil differences would enrich the manuscript. Although the authors note the lack of directional hypotheses, a non-zero contrast was presumably anticipated to enable correlational analyses. The authors' prior response indicated that the contrast's direction might vary with sample composition and misophonia severity; elaborating on expected patterns (in either direction), their potential replicability, and their implications for sound processing could be valuable. For instance, misophonic triggers may have received more attention, due to some of the differences from generally aversive sounds the authors mentioned. I would welcome the authors' perspective on these possibilities, given my limited expertise in this domain.

Minor points

Thank you for clarifying the rating scale definitions, which now appear more ordinal than previously described. I remain interested in the rationale for the response input mode (three adjacent keys with a physically offset key for the highest rating). Additionally, while pupillometry offers advantages over categorical self-reports for assessing misophonia symptom severity, could a visual analogue scale achieve comparable precision (e.g., in computerized diagnostic settings)?

Regarding participant instructions, I question whether explicitly directing participants not to blink during sound presentation would have been advisable. Since the authors' data show rather average blink rates suggesting natural behavior, such an instruction might have imposed unintended inhibitory demands, potentially confounding the task. In my opinion, it was thus appropriate not to include such an instruction.

Reviewer #2: I have carefully reviewed the revised manuscript. The authors have adequately addressed my previous comments, and I am satisfied that the scientific revisions have been implemented appropriately. At this stage, I have only one minor comment and one editorial note.

Minor comment

In the clean copy (Lines 324–327), the newly added description of the phenomenon is interesting. It may be related to the report in Ref. [76], which suggests that, beyond the conventionally known arousal enhancement on the order of ~3 seconds, there may also be a component of enhancement that persists for a longer duration, around ~10 seconds, in a more baseline-like manner.

Editorial note

I reviewed the tracked-changes version of the manuscript; however, it appears that the change-tracking may have been generated via an automatic post-hoc document comparison. As a result, some passages that are textually identical are shown as modified (in red), and there are also instances where the layout/formatting appears disrupted. Therefore, I was not able to reliably confirm whether all fine formatting issues are fully resolved in the revised version. I strongly recommend that the authors carefully check the final clean manuscript themselves and ensure, during the proofing process, that there are no unintended formatting or layout errors.

7. PLOS authors have the option to publish the peer review history of their article (what does this mean?). If published, this will include your full peer review and any attached files.

Reviewer #1: **Yes:**Jens Kürten

Reviewer #2: No

---

## [Editor Report · Decision Letter 2]

6 Apr 2026

PONE-D-25-31481R2Sound-evoked pupil dilation quantifies misophonic symptomsPLOS One

Dear Dr. de Gee,

Thank you for submitting your manuscript to PLOS ONE. After careful consideration, we feel that it has merit but does not fully meet PLOS ONE’s publication criteria as it currently stands. Therefore, we invite you to submit a revised version of the manuscript that addresses the points raised during the review process. **Editor comment.** I have received feeback from the two reviewers that commented previously, and both found that the manuscript has improved, while still requiring further revision. I give an overview of what I think the reviewer mean to address: R1 argues that several claims in the manuscript are based on difference scores (typically low in reliability) which require addressing because the interpretability of any correlation depends on the reliability of the measures involved. R1 therefore requests that appropriate reliability estimates (e.g., split-half or test–retest) be provided, and that the implications of this reliability for the interpretation of the correlations be clarified in the discussion. R2 argues that because the study uses pupil size as a central measure; pupil size is influenced by general arousal, which comprises both phasic and tonic components, and both components can affect performance. Under these conditions, pupil-based effects may reflect general arousal, and this again is critical to the question of what target construct is under study, which potentially could introduces a confound in the interpretation of the findings. R2 therefore requests that this issue be addressed at a theoretical level. I thereby ask you to provide a revision of the manuscript that addresses these remaining concerns in detail, together with a cover letter that addresses each of the reviewers comments in a point-by-point reply manner. Best wishes and good luck with the revision!

We look forward to receiving your revised manuscript.

Kind regards,

Michael B. Steinborn, PhD

Section Editor

PLOS One
---

## [Editor Report · Decision Letter 3]

14 Apr 2026

Sound-evoked pupil dilation quantifies misophonic symptoms

PONE-D-25-31481R3

Dear Dr. de Gee,

We’re pleased to inform you that your manuscript has been judged scientifically suitable for publication and will be formally accepted for publication once it meets all outstanding technical requirements.

I have now received your revised and finalised manuscript, and it appears to be well prepared. The points raised in the previous round of review are addressed adequately, and on this basis, I am satisfied that the concerns raised by the reviewers have been resolved. I am therefore pleased to inform you that your manuscript is accepted for publication.

But there is only a minor issue that I noticed incidentally. Some DOIs appear to be missing or incomplete in the reference list. I would ask you to check and complete these during the final preparation of the manuscript, this is a 2 secs task and can be handled at the production stage without further review.

Apart from this small point, the manuscript is in excellent shape. Congratulations on the successful revision!

Best regards,

Michael B. Steinborn, PhD

Section Editor

PLOS One

Kind regards,

Michael B. Steinborn, PhD

Section Editor

PLOS One

Additional Editor Comments:

remaining task is to check the reference list :

de Gee, J. W., Colizoli, O., Kloosterman, N. A., Knapen, T., Nieuwenhuis, S., & Donner, T. H. (2017). Dynamic modulation of decision biases by brainstem arousal systems. eLife, 6, e309. https://doi.org/10.7554/eLife.23232

Joshi, S., Li, Y., Kalwani, R. M., & Gold, J. I. (2016). Relationships between pupil diameter and neuronal activity in the locus coeruleus, colliculi, and cingulate cortex. Neuron, 89, 221-234. https://doi.org/10.1016/j.neuron.2015.11.028

Knapen, T., de Gee, J. W., Brascamp, J., Nuiten, S., Hoppenbrouwers, S., & Theeuwes, J. (2016). Cognitive and ocular factors jointly determine pupil responses under equiluminance. PLOS ONE, 11, e0155574. https://doi.org/10.1371/journal.pone.0155574

Murphy, P. R., O'Connell, R. G., O'Sullivan, M., Robertson, I. H., & Balsters, J. H. (2014). Pupil diameter covaries with BOLD activity in human locus coeruleus. Human Brain Mapping, 35, 4140-4154. https://doi.org/10.1002/hbm.22466

Reimer, J., McGinley, M. J., Liu, Y., Rodenkirch, C., Wang, Q., McCormick, D. A., et al. (2016). Pupil fluctuations track rapid changes in adrenergic and cholinergic activity in cortex. Nature Communications, 7, 13289. https://doi.org/10.1038/ncomms13289

Thiele, A., & Bellgrove, M. A. (2018). Neuromodulation of attention. Neuron, 97, 769-785. https://doi.org/10.1016/j.neuron.2018.01.008
---

## [Editor Report · Acceptance letter]

PONE-D-25-31481R3

PLOS One

Dear Dr. de Gee,

I'm pleased to inform you that your manuscript has been deemed suitable for publication in PLOS One. Congratulations! Your manuscript is now being handed over to our production team.

Kind regards,

on behalf of

Dr. Michael B. Steinborn

Section Editor

PLOS One